# Preliminary Investigation into Developing the Use of Swabs for Skin Cortisol Analysis for the Ocean Sunfish (*Mola mola*)

**DOI:** 10.3390/ani12202868

**Published:** 2022-10-21

**Authors:** Rachel M. Santymire, Marissa Young, Erin Lenihan, Michael J. Murray

**Affiliations:** 1Biology Department, Georgia State University, 100 Piedmont Ave SE, 4th Floor, Atlanta, GA 30303, USA; 2Veterinary Services, Monterey Bay Aquarium, 886 Cannery Row, Monterey, CA 93940, USA

**Keywords:** acclimation, fish, euthanasia, glucocorticoid, illness, injury, stress physiology

## Abstract

**Simple Summary:**

Zoos and aquaria play an important role in preventing the mass extinction of wildlife through public awareness of conservation issues and providing a safe haven for wildlife populations. Because aquatic populations face many challenges due to pollution and global warming, it is important to develop an understanding of how species can cope with their environment whether it be in the wild or under human care. Here, we were interested in developing non-invasive methods to study fish stress physiology. We use the unique ocean sunfish (*Mola mola*) to develop the use of skin swabs to measure the stress hormone, cortisol. We used known times of stress including when a mola was injured or ill and during acclimation to the Monterey Bay Aquarium. We found that cortisol increased initially within the first month of being admitted to the aquarium, but returned to normal values afterward. Molas also had elevated cortisol when being treated for an injury or illness. This is the first step in the development of the use of skin swabs to collect samples for stress analysis in the mola. Additional biochemical analysis is needed to confirm these results and allow this method to be applied to other species of fish.

**Abstract:**

The ocean sunfish (mola; *Mola mola*) is the heaviest bony fish in the world. This slow-moving fish often is injured by fishing boats that use drift gillnets attributing to its listing as Vulnerable by the IUCN. The Monterey Bay Aquarium (Monterey, CA, USA) has a program that brings in smaller molas from the ocean and acclimates them for an exhibit. When they grow too large for the million-gallon Open Seas exhibit, they are returned to Monterey Bay through a “reverse” acclimatization. Our overall goal was to use skin swabs to evaluate mola stress physiology to better understand the effects of this program. Our objectives were to validate this non-invasive method by taking opportunistic swabs throughout acclimatization and during stressful events. We swabbed each individual (*n* = 12) in three different body locations. Swabs were analyzed using a cortisol enzyme immunoassay. We averaged the three swabs and examined the absolute change of cortisol from the first taken upon handling to during treatments and the different acclimation stages. We considered elevated cortisol concentrations to be ≥1.5-fold higher than the first sample. Overall, mean (±SEM) cortisol varied among individuals (564.2 ± 191.5 pg/mL swab (range, 18.3–7012.0 pg/mL swab). The majority (four of six) of molas swabbed within the first week or month had elevated skin cortisol compared to their first sample. All seven molas that were being treated for an injury or illness had elevated skin cortisol (range, 1.7- to 127.6-fold higher) compared to their post-acclimation sample. This is the first step in validating the use of non-invasive skin swabs for glucocorticoid analysis in the mola. Further biochemical analysis is needed to determine the specific steroids that are being measured.

## 1. Introduction

As humans continue to alter the environment and use its natural resources, wildlife habitat and populations have become negatively impacted with an estimated one million species at risk of extinction [1]. Additionally, much anthropogenic activity has increased the levels of atmospheric carbon dioxide. Our oceans sequester a considerable amount of this carbon, which results in ocean acidification and has negatively affected marine biodiversity [2]. Because aquatic populations are vulnerable to extinction and face a stochastic environment, it is important to develop an understanding of how species can cope physiologically with their environment.

One method of determining how the environment has affected wildlife health is to evaluate stress physiology. In vertebrates, adrenocortical hormones, called glucocorticoids (GCs), are released after a perceived stressor triggers the release of corticotropin-releasing hormone (CRH) from the hypothalamus, which stimulates the release of adrenocorticotropin hormone (ACTH) from the anterior pituitary. The ACTH, then, acts on the adrenal glands (or interrenals in other species like in amphibians and teleost fish) to release GCs (i.e., Hypothalamic-Anterior Pituitary-Adrenal or Interrenals axis, HPA or HPI) [3,4]. The GCs mobilize energy to allow the individual to cope with the stressor, but also for several other biological factors including thermoregulation, nutritional state, and circadian rhythm (reviewed in [3,5]). Responses to acute stressors are adaptive and often assist individuals with coping with their environment as observed in fish (reviewed in [5,6,7]. However, repeated or chronic HPA stimulation can negatively impact animal health. For example, chronic gluconeogenesis results in hepatocellular degeneration [4]. Behavioral and cognitive abilities are also negatively altered along with immunosuppression, which can increase one’s vulnerability to disease [8,9,10,11]. Finally, GCs reduce the stimulation of the HP-Gonadal (HPG) axis which ultimately affects reproduction by decreasing the production of gonadal steroids, like estrogen, progesterone, and testosterone [12]. Therefore, chronic GC production can ultimately lead to population declines [13,14].

In mammals and birds, evaluating GC production traditionally involved measuring cortisol or corticosterone in blood, feces, urine, and saliva [15,16,17,18]. All of these sample types have their advantages and disadvantages. GC analysis in blood requires rapidly capturing the animal and drawing blood to avoid an increase in these hormones due to the stress of capture [19,20]; however, blood allows for the measure of the direct quantification of the steroid hormone itself and not a hormonal metabolite, which is present in feces and urine. Analysis of GC metabolites requires validation to demonstrate that the metabolites measured are biologically relevant to that species’ physiology [17,18]. More recently, GCs have been quantified in hair and feather samples, but this method also requires additional validation steps (reviewed in [21,22,23]). Most of these samples are difficult or impossible to collect on non-mammalian, non-avian and/or semi- and fully aquatic species (because they do not have hair, for example). Some researchers have overcome the challenge of studying the endocrinology of whales by using blow [24,25,26] and feces [27]. However, even with these types of sampling, researchers need to be in the right place at the right time.

Typically, the aquatic environment also makes it difficult to collect urine and feces (see Figure 1 in [28]). In amphibians, researchers have analyzed hormones using a water container and quantified the concentration of GCs in a known amount of water [29,30,31,32]. Additionally, GCs have been collected using dermal swabs and have been validated for several amphibian species [33,34,35]. More recently, dermal swabs have been used to study the stress physiology of three cephalopod species (*Euprymna berryi*, *Sepia bandensis*, and *Octopus chierchiae* [36]). In this study, swabs were used underwater to collect mucus from the cephalopods, which the authors used to measure GCs. The authors also used an ACTH challenge and acclimation to a new habitat to validate GCs analysis [36].

Similar to amphibians, methods for evaluating steroid hormones in teleost fishes have included measuring hormones in water from a tank in which the individual is placed (Reviewed by [37]) and blood, which may require sedation or euthanasia [38]. In fishes, voided urine and feces are often lost and diluted in the water [39]. Unlike amphibians, fish scales have been used to evaluate steroid hormones. Similar to feathers and hair, scales generally continue to grow throughout life [40]. It is hypothesized that GCs are deposited in the scales via blood similar to feathers [23] in birds and hair in mammalian species [21,22]. Another hormonal sampling method shared by fishes and amphibians is collecting skin secretions. However, in fishes, mucus is collected using a cell scraper [38,41] versus a cotton swab used in amphibians [33,34,35] and cephalopods [36].

Our goal was to study the stress physiology of the ocean sunfish (mola; *Mola mola*) using skin mucus collected non-invasively using swabs. The mola is the heaviest bony fish in the world capable of reaching 333 cm total length and 2.3 tons in size [42]. Despite its small, lateral fins and truncated tail (clavis), it is capable of rapid movement and even adult fish have been observed leaping out of the water [43]. The mola’s habit of basking in lateral recumbency at the surface of the water makes it vulnerable to boat strikes and attacks by gulls and pinnipeds. The mola is listed as Vulnerable by the IUCN Redlist [42]. While the species is not targeted for human consumption, it is endangered by bycatch. The IUCN considers the population trend to be decreasing.

The mola is not commonly managed as an exhibit fish in aquariums. Its size and rapid growth rate make it impractical for all but institutions with large tanks to properly house and care for these charismatic fish. The mola seems to have some idiosyncratic behavioral responses to what might be considered rather innocuous circumstances. Unanticipated approach or contact by other large animals, such as tuna, sea turtles, and sharks, co-inhabiting the sunfish’s tank often results in abnormal behavior, such as inappetence, swimming very close to the wall instead of in the midst of the water column, and repeatedly swimming into the tanks walls, bottom, and acrylic viewing window.

We elected to take advantage of the visible shifts between low and high-stress behaviors, the robust production of skin mucus in the species, and the operant training of the mola to approach a target and subsequent handling within a vinyl stretcher to investigate the GC-based stress response in the mola. Our objectives were to: (1) use non-invasive skin swabs to evaluate mola stress physiology by taking opportunistic swabs throughout acclimation to the aquarium; and (2) determine if changes in GC production can be detected during stressful events, such as the behavioral anomalies described above, illness, injury, and euthanasia. When performed, euthanasia methods were compliant with the most current version of the AVMA Guidelines, and generally were conducted using an overdose of MS-222.

## 2. Materials and Methods

### 2.1. Study Area

All aspects of this research were approved by the Lincoln Park Zoo Research Committee and IACUC (#2017-012) and by the Monterey Bay Aquarium Research Oversight Committee. Molas were maintained at the Monterey Bay Aquarium (Monterey, CA, USA) in one of three locations: the 1.3 million gallon Open Seas Exhibit with other pelagic fishes or one of two off-site holding tanks measuring 30 or 40 ft in diameter. The tanks contain filtered, natural seawater with a target temperature of 21–22 °C.

### 2.2. Sample Collection

Subject fish were called to the surface using an individual-specific target. Once at the surface, a vinyl stretcher was used to restrain the mola as it was lifted out of the water. Samples were taken immediately with time ranging from 1 to 5 min. Meanwhile, other staff members were covering the eyes with a wet chamois cloth. The animal was ventilated using a Nalgene tube connected to a submersible pump. The tube was inserted into the mouth and water was pumped over the gills and out through the opercula.

We swabbed each individual (*n* = 12) between 11:00 and 13:00 and in three different body locations: either the left or right body wall, the clavis, and either the anal or dorsal fin. Care was taken to ensure that the sampled body part was not submerged and that seawater was not coursing over the skin at the time of sampling. We swabbed back and forward three times spanning 1 inch using a swab with a detachable end (OmniSwab Sterile, Whatman Inc, Clifton, NJ, USA). Swabs were taken opportunistically during different stages of acclimation from the admit examination upon capture from Monterey Bay to within 1, 4, 5, 6, and 7 weeks post-capture. Individuals were also sampled when there were health issues, such as injury, signs of illness, and prior to euthanasia due to one of these life events. Finally, on three occasions, molas were swabbed immediately upon examination and then 15 min later to determine how quickly cortisol increases in the mucus upon capture. Swabs were immediately placed in individually labeled 2.0 mL tubes containing 1 mL of 70% ethanol and, then, stored at −80 °C until they were shipped to the Lincoln Park Zoo (Chicago, IL, USA) for processing and analysis.

### 2.3. Sample Processing and Analysis

Using previously described methods from Santymire and colleagues [26], the vials of the 1 mL of 70% ethanol containing the swab were mixed (Glas-col, Terre Haute, IN, USA; setting 60–70 rpm) for 5 min and, then, 500 µL of 70% ethanol was placed into a clean test tube. The ethanol was evaporated using forced air in a warm water bath (60 °C). To reconstitute the samples, 500 µL of phosphate-buffered saline (PBS) was added along with two to three glass beads, which aided in removing the samples off the sides of the test tubes. The samples were briefly vortexed and sonicated for 20 min. Samples were mixed again on the Glas-col mixer (60–70 rpm) for 30 min as the final step before being analyzed on a cortisol enzyme immunoassay (EIA).

We used a polyclonal antiserum (R4866) and horseradish peroxidase (HRP) for cortisol (provided by C. Munro, Davis, CA, USA) diluted at 1:375,000 and 1:200,000, respectively. We used a double-antibody enzyme immunoassay system on microtiter plates pre-coated with a secondary goat anti-rabbit IgG (Arbor Assays, Ann Arbor, MI, USA) using methods described by Edwards and colleagues [44]. For the cortisol antiserum cross-reactivity see Munro and Stabenfeldt [45].

For the biochemical validation of the cortisol EIA, a parallelism between binding inhibition curves of sample hormones (2× concentrated to 1:4) and the cortisol standard (*r* = 0.965) was evaluated along with the EIA producing a significant recovery (≥90%) of exogenous cortisol (1.95–500 pg/well) added to a pooled swab sample (y = 1.08x − 2.38; R² = 0.999; *p* < 0.001). Assay sensitivity was 1.95 pg/well. The intra-assay coefficient of variation (CV) was <10% and the inter-assay CV was 7.3%.

### 2.4. Statistical Analysis

To determine the amount of change in skin cortisol during stress events, we compared the absolute change (i.e., the fold increase [46]) by calculating the proportion between the first sample taken and samples during what we considered stressful events [40]. For example, a 1-fold change value indicates no change compared to the initial value. Therefore, we considered skin cortisol to be elevated if the value was ≥1.5 fold higher than the first sample and an indication of a stress response [33]. To determine if the change in skin cortisol varied significantly (*p* < 0.05), we first tested skin cortisol for normality using Shapiro–Wilk for normality assumption testing and Levene’s median test for equal variance assumption (Sigma Plot version 12, San Jose, CA, USA). Data were normal; therefore, we used the parametric tests. Because we had few individuals within each event, we used statistical methods that compared data within an individual. To determine if skin cortisol statistically increased within 15 mins post-capture (n = 3 individuals), we compared the first sample upon capture to the sample taken 15 mins later using a paired *t*-test. To determine if skin cortisol changed over time during acclimation to the aquarium (n = 5 individuals), we used one-way repeated measures ANOVA. Finally, we used paired *t*-tests to determine if an individual’s skin cortisol increased during the treatment of an illness or injury and euthanasia.

## 3. Results

### 3.1. Skin Cortisol Variability

In total, 51 samples, in triplet, were collected from the 12 molas with a mean of 4.3 ± 0.8 samples per mola (range, 2 to 11 samples). The triplicates (taken from three different areas on each individual) of each time point were highly varied with a mean percent Coefficient of Variance (CV) of 59.3 ± 4.4% (range, 7.6 to 161.4%). Unfortunately, the location of each sample was not documented on the sample vial; therefore, we could not compare sites. So, we averaged the three samples at each time point. The overall mean skin cortisol value for the 51 samples was 564.2 ± 191.5 pg/mL swab (range, 18.3–7012.0 pg/mL swab). Additionally, from time 0 mins to 15 mins post-capture, we observed an increase in skin cortisol in the three molas from 1.1-, 1.4-, and 1.8-fold-increase.

### 3.2. Acclimation to the Aquarium

To determine the effects of the different stages and time frames of acclimation, we had five individuals with samples collected immediately (within 5 mins) upon capture, within the first week in the aquarium holding tanks and 4, 8, 12, 20, 24, and 28 weeks post-capture (Figure 1). The responses were similar (F_4,20_ = 1.615; *p* = 0.244) across the different time points of acclimation in the five individuals. Two out of three individuals that were swabbed within the first week of admittance to the aquarium had elevated (1.5-fold or greater) skin cortisol (Mola 16-07, 1.5-fold; Mola 16-09, 2.2-fold higher) compared to their first sample. Two out of three molas that were swabbed after the first week, but within the first month had elevated skin cortisol (Mola 17-04, 3.7-fold; Mola 17-05, 1.6-fold) compared to their initial sample. In all other samples after the first month (from 2 to 7 months) post-capture, skin cortisol values were not elevated sample (range, 0.04- to 0.6-fold increase) compared to the initial. When we compared the first to the second samples (either week one or four) taken during the acclimation time, we found a near significant increase (t_4_ = −2.41; *p* = 0.074).

### 3.3. Effects of Health Status on Skin Cortisol

We had seven individual molas with a sample taken post-acclimation prior to being injured, becoming ill, or having to be euthanized due to health reasons (n = 16 samples total; range, 2 to 6 per mola). The mean fold-increase in skin cortisol due to health issues was 30.3 ± 9.8-fold increase with a range of 1.7 to 127.6-fold higher than the initial sample. For all of the responses to illness, skin cortisol did not vary (t_2_ = −2.59; *p* = 0.123). Specifically, when Mola 16-08’s health started declining, it had a 127.6-fold-increase in skin cortisol. Three months later, its health had returned to normal and its skin cortisol was back to post-acclimation value (Figure 2A). Mola 16-09 had a 110-fold increase in skin cortisol after being injured from a collision with tuna in the exhibit (4764.0 pg/mL swab) compared to its post-acclimation sample (43.7 pg/mL swab). Another mola (17-04) had a 50-fold increase in skin cortisol when it started receiving treatment for a debrided dorsal fin (Figure 2B).

During the time frame of this opportunistic study, four molas had to be euthanized due to failing health or injury. Mola 16-05 had an 18.7-fold increase in skin cortisol when it started showing deteriorating behavior. Then, 9 days later, its skin cortisol had increased 38.2-fold higher than its post-acclimation sample when it had to be euthanized (Figure 2C). Mola 16-06 had 38.1-fold higher skin cortisol when it was euthanized due to an injury sustained after becoming entrapped in the tank lining (5470.1 pg/mL swab) compared to its post-acclimation sample (143.7 pg/mL swab). When Mola 18-01 was found with large abrasions (197.6 pg/mL swab), its skin cortisol was 6.4-fold higher than its post-acclimated sample (30.7 pg/mL swab), and the veterinary staff decided to euthanize it. Finally, Mola 18-03 had a persistent growth of masses on its sternum and pectoral and anal fins while in the aquarium for 6 months. Its skin cortisol had increased 2.8-fold from 18.3 to 51.7 pg/mL swab. After it was found entrapped on exhibit with numerous scrapes, it was euthanized with a 59.3-fold increase in skin cortisol (1084.3 pg/mL swab). In all of these individuals, euthanasia did not result in higher (t_2_ = −2.34; *p* = 0.144) skin cortisol value compared with its post-acclimation sample.

## 4. Discussion

Zoos and aquaria play an important role in the conservation of wildlife and their habitats by presenting wildlife to the public using modern zoological practices for the benefit of the individual animals who are acting as ambassadors for their species. It is incumbent upon those responsible for the care of these animals, and a requirement of the Association of Zoos and Aquariums (AZA) accrediting body, that animal welfare remains of paramount importance. Within the zoo and aquarium profession, animal welfare embraces the five opportunities for animals to thrive. One of these tenets is to allow animals to avoid chronic stress.

Our goal was to take the first steps in validating the use of skin mucus swabs as a non-invasive tool to study fish stress physiology using the mola as a model species. While using mucus from the skin of fish to study their physiology is not a novel method [47], using the less invasive approach of a swab instead of a cell scrapper may serve as an improved technique for collecting a standardized amount of mucus since it has been suggested that scraping of the mucus may compromise the protective nature of the integument [39].

As mentioned previously, it is important to ensure that the same technique is used to collect the mucus by standardizing the location and the procedure of swabbing (here, up and back 1 inch three times). We found high variability among the three samples taken at each time point. Unfortunately, we could not compare skin cortisol concentrations among body sites since that information was not recorded; therefore, we had to average the three samples per time point. Different rates of GC secretion have been found between body locations (dorsal versus ventral) in the edible bullfrog (*Pyxicephalus edulis* [35]). Additionally, these authors found slightly higher dermal GC concentrations in the ventral samples compared to the dorsal samples. In the Senegalese sole (*Solea senegalensis*), skin cortisol was similar between the dorsal and ventral sides [48].

We also observed variability in the initial skin cortisol values. During the admittance examination, mola skin cortisol ranged from 50 to ~500 pg/mL swab. Interestingly, we found that skin cortisol values for the acclimated molas that were injured or ill was ~30 pg/mL swab (range, 18.3–143.7 pg/mL swab). This variability could be attributed to handling time before collecting the swab and/or based on each individual’s experience with handling. It has been shown that individual stress responses are influenced by several factors including experience, age, food availability, perceived risks, and health status [49]. Variability in skin mucus cortisol has been observed in other species. For example, the gilthead seabream (*Sparus aurata* L.) had cortisol concentrations found in skin mucus that ranged from 3.3 to 112 ng/mL mucus [41]. Similarly, the Catalan chub (*Squalius laietanus*) had a range of skin cortisol of 20 to 80 ng/mg of mucus [38]. Fernandez-Alacid and colleagues [50] compared mucus cortisol in three species meagre (*Argyrosomus regius*), European sea bass (*Dicentrarchus labrax*), and gilthead seabeam, and found that values ranged from ~1 to 17 ng/mL mucus. All of these papers have skin mucus cortisol concentrations in the ng/mg mucus (or ng/mL mucus). Our values were in the pg/mL swab concentrations, which are not directly comparable. This might be due to the fact that we are not collecting the same amount of mucus that these authors collected. However, these skin cortisol concentrations are similar to other species including amphibians [33,34].

It has been suggested that skin cortisol may reflect circulating hormones quickly with minimal lag time [33]. In blood, Lawrence and co-authors [20] demonstrated that the lag time between a stressor and GC circulating in the blood was within 4 to 8 min in teleost fish. Interestingly, we did observe a rapid increase in skin cortisol after capture with one out of three molas having elevated (≥1.5 fold-increase) cortisol within 15 min. A rapid increase in dermal cortisol or similar glucocorticoid was also observed in amphibians including the Wyoming toad (*Anaxyrus baxteri*; [34]), three species of cephalopods [36], American toad (*Anaxyrus americanus*), red-spotted newt (*Notophthalamus viridescens*; [33]) using the cotton swabs. In fish, DeMercado and colleagues [51] found that cortisol increased in the skin mucus of rainbow trout (*Oncorhynchus mykiss*) as early as 10 min upon capture. Most of these authors did not resample individuals under 30 to 60 min [38,50,52,53]. Our data suggest that it is important to sample quickly upon capture and handling to be able to accurately evaluate fish stress physiology.

Although skin cortisol varied among individuals, we did observe an increase during stressful events in all individuals. GC analysis has been validated using physiological testing, such as an ACTH challenge, and using biological events including transport, translocation, social stressors, veterinary procedures, and environmental conditions (reviewed in [18]). Here, we used known stressors (acclimation to a new environment, injury, and illness) to evaluate the validity of using non-invasive skin swabs to evaluate mola stress physiology. Acclimation to a new environment is a proven stressor in cephalopods [36,54,55,56] and mammalian species [16,57]. Here, we did not find that the acclimation process significantly increase skin cortisol; however, we found that within the first week two of three individuals’ skin cortisol was elevated (≥1.5 fold here than initial sample). Similarly, two of three individuals had elevated skin cortisol within 2 to 4 weeks. Unfortunately, none of the individuals was measured in both the first week and month, so we do not exactly how long it does take them to acclimate; however, all five individuals had low skin cortisol (i.e., near post-acclimation values) after the first 4 weeks and up through 7 months in the new habitat. Therefore, our data suggest that it may take up to 2 months for the individuals to acclimate to their new habitat.

In addition to acclimation to a new environment, injury and illness can induce a stress response (via an increase in cortisol). Narayan and colleagues [58] found increases in fecal cortisol metabolites in the greater bilby (*Macrotis lagotis*) after injury and illness. In three-banded armadillos (*Tolypeutes matacus*), veterinary procedures evoked stress responses which were demonstrated by an increase in fecal cortisol metabolites [59]. Although we did not find that skin cortisol was significantly higher in mola with injuries that did or did not lead to euthanasia, we did observe the greatest increase in skin cortisol (127.6-fold increase) in a mola with declining health. In another mola, skin cortisol increased ~110-fold after trauma from a collision with tuna. Finally, we also observed ~40 to 60-fold increases when individuals had to be euthanized due to declining health and trauma.

## 5. Conclusions

Our goal was to initiate the validation of using skin swabs to collect fish mucus for GC analysis. Developing less invasive techniques to improve our understanding of fish stress physiology would improve fish *ex situ* management and conservation. Here, we show that molas tended to have declining skin cortisol by 8 weeks post-capture suggesting that they acclimated to the aquarium. We have biologically validated the use of skin swabs for collecting fish mucus for hormonal analysis using data from injured and ill molas. However, further biochemical analysis is needed to determine the specific steroids that are being measured in the mucus from the skin so that these methods can be applied to other fish species. Future research should standardize the sample collection protocol by body location, time of day, and minimizing handling time to ensure the resulting skin cortisol data is biologically relevant to each individual. By understanding fish stress physiology, we can better understand how their environment, both natural and human-managed care, is impacting their health.

## Figures and Tables

**Figure 1 animals-12-02868-f001:**
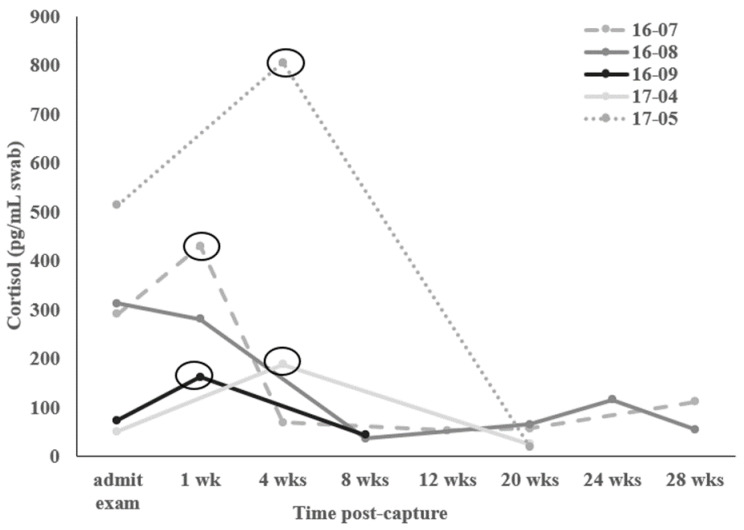
Skin mucus cortisol values (pg/mL swab) from admittance examination upon capture from the Monterey Bay (Monterey, CA, USA) through 28 weeks post-capture for five mola individuals. Circles indicate an elevated cortisol value (≥1.5-fold increase [33]) compared to the first, initial sample taken during the admittance exam.

**Figure 2 animals-12-02868-f002:**
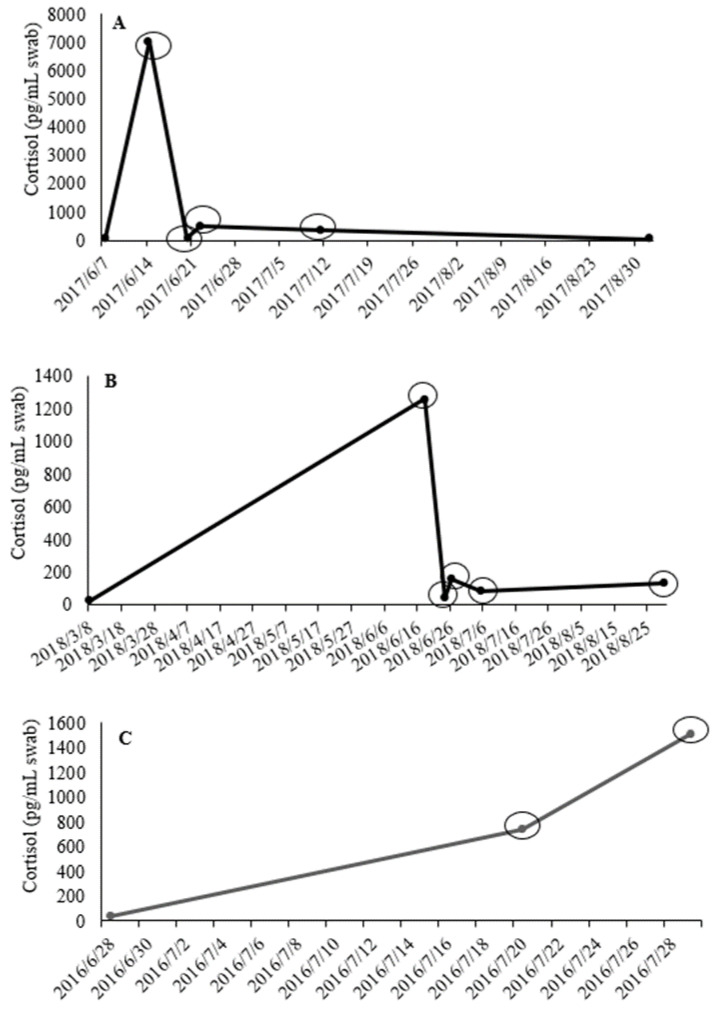
Skin mucus cortisol values (pg/mL swab) from a post-acclimation sample during when the molas were considered healthy to time of illness and recovery (**A**), injury (**B**), and signs of illness that lead to euthanasia (**C**). Circles indicate an elevated cortisol value (≥1.5-fold increase [33]) compared to the post-acclimation sample.

## Data Availability

These data are available per request made to the corresponding author, R. Santymire.

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
