# Peer review of "Preliminary Investigation into Developing the Use of Swabs for Skin Cortisol Analysis for the Ocean Sunfish (Mola mola)"

_animals, 2022, doi:10.3390/ani12202868_

Round 1

Reviewer 1 Report

Review of Santymire et al. "Preliminary investigation into developing the use of swabs for skin cortisol analysis for the ocean sunfish (Mola mola)".

Manuscript ID: ANIMALS-1871018

This study seeks to develop a simple and non-invasive methodology to assess stress in a teleost fish, Mola mola. Through the use of swabs, the authors collect samples of the dermal mucus and analyse cortisol. It is a novel study in that it applies existing knowledge to a protected species that is kept in aquariums for public display. The text is wonderfully written and abounds in comparative physiology references. Nevertheless, the Introduction section in mostly focused in non-fish species. The authors are encouraged to increase the number of references related to teleosts, as stress responses may differ from fish to tetrapods.

One of the weakest points of this work is the lack of a statistic that helps to make the results more robust. Authors are encouraged to contact an expert in statistics to improve the work. Due to this, and the lack of a better description of the methodological processes used, this study could be published after major changes.

Some specific concepts that should be considered are detailed below:

INTRODUCTION

- Lines 54-55: glucocorticoids are released in vertebrates. This should be clarified in the text.

- Line 57: the “adrenal glands” in teleost fish are called “interrenal cells”, due to their morphological differences with those in tetrapods.

- Lines 61 and 65: All these references are from mammals. The authors are encouraged to include studies conducted in fish, with special emphasis on teleosts (as Mola mola is a teleost). There is plenty of information on this topic.

- Line 71: Again, it would be much better to include information from teleosts or other fish. Some examples are (in chronological order):

·         Aerts et al., 2015. DOI: 10.1371/journal.pone.0123411. Glucocorticoids (cortisol) measured in scales.

·         Fernández-Alacid et al., 2018. DOI: j.scitotenv.2018.07.083. A reference paper where dermal mucus served in teleosts to measure stress responses.

·         Skrzynska et al., 2018. DOI: 10.3389/fphys.2018.00096. A comprehensive study of some primary and secondary stress responses in teleost fish.

·         Fernández-Alacid et al., 2019. DOI: 10.1016/j.aquaculture.2018.09.039. Relationships between plasma and skin mucus cortisol in a teleost fish.

- Line 74: Again, it will be better to include “fish-related” references. For example:

·         Lawrence et al., 2018. DOI: 10.1139/cjz-2017-0093.

- Line 90: Just curious: Do the authors really consider that cephalopods produce steroid molecules?

MATERIAL AND METHODS

- Lines 143 – 150: What was the time from the moment the fish perceived they were being lifted from the water, until the samples were collected? According to some cited references, this time should be less than 3 minutes to assess stress responses different than those due to fish handling. Sampling is, thus, of major importance in the present study. Please, describe precisely everything related to this procedure.

- Line 179, Statistics: If the authors wish to publish the results in a scientific journal (since this study seems useful for the scientific community), they should make a statistical effort to strengthen their conclusions. Perhaps talking to a statistician will help to this end.

RESULTS

- Lines 191-192: this first sentence makes no sense, as the number of samples will depend on some independent variables the authors may wish to include in the statistical analysis.

- Line 192: what do the authors refer to “triplicates”? In M&M, they stated samples were taken from 3 different body areas from each fish. Are these the triplicates? Please, clarify.

- Line 193: The deviation from the mean seems too high to ensure a proper methodological procedure.

- Line 194: the location of each sample should be documented on the sample vial. This is mandatory if the authors want to publish these results.

- Line 196: Do the authors measured the volume of each swab? That information is not described in M&Ms. Please, include that information.

- Lines 196 – 198: Statistics are required.

- Line 202: what exact time point do the authors consider as “immediately upon capture”? This information is relevant and the capture process should be also explained.

- Line 206: The referred “baseline sample” should be described through statistical methodologies.

- Figure 1: Do the authors considered that stressed animals produce more dermal mucus than under normal circumstances? This could dilute the cortisol in the mucus. Moreover, the authors did not measured the volume of skin mucus collected, so that they can’t show the results in “mL swab”.

-  Lines 217-219: the illness and euthanize protocols should be describe to better understand how this study was conducted.

- Figure 2: The authors can´t link all sampling points with lines, as they measured cortisol at discrete times, not continuous. Moreover, since cortisol in teleost fish has a marked circadian rhythm, it is extremely important to show the time of day at which all samples were collected.

 DISCUSSION

- Line 273: This is the first time the exact cortisol concentration per swab is shown in the manuscript. This information should be placed in the Results section.

- Line 282: “seabream” instead of “sea bream”.

- Line 284: The results from the present study can´t be directly compared to those from the cited references as they measured the volume of collected mucus.

- Line 287: There is ample evidence that cephalopods do not produce cortisol… Please, revise the literature.

- Lines 289-290: Please, refer to studies conducted in fish.

Author Response

Thank you for your time and effort in our paper. The paper is much stronger and more specific to teleost fish thanks to these helpful reviews. Below we have added our responses to your questions in bold.

INTRODUCTION

- Lines 54-55: glucocorticoids are released in vertebrates. This should be clarified in the text.

We have added “In vertebrates” for clarification.

- Line 57: the “adrenal glands” in teleost fish are called “interrenal cells”, due to their morphological differences with those in tetrapods.

We have added “(or interrenal cells like in teleost fish)

- Lines 61 and 65: All these references are from mammals. The authors are encouraged to include studies conducted in fish, with special emphasis on teleosts (as Mola mola is a teleost). There is plenty of information on this topic.

We have added some “the effects of “stress” on teleost fish to this part of the introduction and included HPI.

- Line 71: Again, it would be much better to include information from teleosts or other fish. Some examples are (in chronological order):

We thank the reviewers for these references that we had not included and have added some teleost fish references to the effect of GC Lines 61-71.

  • Aerts et al., 2015. DOI: 10.1371/journal.pone.0123411. Glucocorticoids (cortisol) measured in scales.
  • Fernández-Alacid et al., 2018. DOI: j.scitotenv.2018.07.083. A reference paper where dermal mucus served in teleosts to measure stress responses.
  • Skrzynska et al., 2018. DOI: 10.3389/fphys.2018.00096. A comprehensive study of some primary and secondary stress responses in teleost fish.
  • Fernández-Alacid et al., 2019. DOI: 10.1016/j.aquaculture.2018.09.039. Relationships between plasma and skin mucus cortisol in a teleost fish.

- Line 74: Again, it will be better to include “fish-related” references. For example:

  • Lawrence et al., 2018. DOI: 10.1139/cjz-2017-0093.

We have added this reference to the manuscript.

- Line 90: Just curious: Do the authors really consider that cephalopods produce steroid molecules?

Yes, hormones more generally, seem to be highly conserved among vertebrates and invertebrates (LaFont, 2000; Ketchesin, et al., 2017; Ottaviani and Franceschi, 1996; Ottaviani, et al., 1992; Adamo, 2012). However, in mollusks, the system may be simplified and contained within the hemocytes, instead of relying on multiple organs as in vertebrates (Ottaviani and Franceschi, 1996). Where glucocorticoids are produced and remain unverified (Ottaviani et al., 1992). 

MATERIAL AND METHODS

- Lines 143 – 150: What was the time from the moment the fish perceived they were being lifted from the water, until the samples were collected? According to some cited references, this time should be less than 3 minutes to assess stress responses different than those due to fish handling. Sampling is, thus, of major importance in the present study. Please, describe precisely everything related to this procedure.

Our goal was to minimize the time between handling and taking the sample. When fish were under human care, the interval ranged from < 1 min to approximately 5 minutes. We have clarified this in the methods.

- Line 179, Statistics: If the authors wish to publish the results in a scientific journal (since this study seems useful for the scientific community), they should make a statistical effort to strengthen their conclusions. Perhaps talking to a statistician will help to this end.

We have added some statistical analyses. Because we had only 3 to 5 individuals per stress event type, we did analyses that evaluated changes within individuals including paired t-tests and one-way repeated measure ANOVA.

RESULTS

- Lines 191-192: this first sentence makes no sense, as the number of samples will depend on some independent variables the authors may wish to include in the statistical analysis.

Our goal in showing the number of samples collected per individual was to make it clear to the readers how much data we are presenting in the paper.

- Line 192: what do the authors refer to “triplicates”? In M&M, they stated samples were taken from 3 different body areas from each fish. Are these the triplicates? Please, clarify.

Yes, triplicates are the three samples that were taken from 3 different locations on each individual during the same time point. We have clarified this in the text.

- Line 193: The deviation from the mean seems too high to ensure a proper methodological procedure.

Again, we wanted our readers to realize that there was a lot of variation so that future studies can adjust their methods to reduce this variation. We thought it would be unethical to randomly choose one sample out of three.  We have emphasized that the location of the sample should be kept consistent to reduce this variation in the Discussion. However, we felt like our results (presented as averaged values) still demonstrate that this method should be considered for future research.

- Line 194: the location of each sample should be documented on the sample vial. This is mandatory if the authors want to publish these results.

Unfortunately, as mentioned above, we don’t have those locations; therefore, we averaged the samples to get an overall mean of skin cortisol.  We still observed high increases in skin cortisol during time of stress.  We also present the mean and CV to be open and honest about our findings.

- Line 196: Do the authors measured the volume of each swab? That information is not described in M&Ms. Please, include that information.

Yes, in Method’s sections 2.2 and 2.3 we describe how the swabs are placed into 1 mL of ethanol. For processing, we take 500 uL of that vial (after mixing it thoroughly). So the value of cortisol is per mL from the vial.

- Lines 196 – 198: Statistics are required.

We have added the statistical results to the Results section.

- Line 202: what exact time point do the authors consider as “immediately upon capture”? This information is relevant and the capture process should be also explained.

We have clarified how quickly samples were taken upon capture both here and in the Methods (within 5 mins).

- Line 206: The referred “baseline sample” should be described through statistical methodologies.

This terminology is confusing since we didn’t statistically find the baseline. Therefore, we have changed the “baseline” to another descriptive language, such as initial sample or post-acclimation.

- Figure 1: Do the authors considered that stressed animals produce more dermal mucus than under normal circumstances? This could dilute the cortisol in the mucus. Moreover, the authors did not measured the volume of skin mucus collected, so that they can’t show the results in “mL swab”.

This was an interesting question; however, Molas don’t increase mucus production during times of stress. As mentioned about, the mL swab is because we had 1 mL of ethanol that we put the swab into. Then we took an aliquot from the vial for analysis.

-  Lines 217-219: the illness and euthanize protocols should be describe to better understand how this study was conducted.

Illness in the captive mola was the result of abrasions associated with rubbing on the tank walls.  The integument of the mola is quite thin and covers a thick, collagenous dermis that resembles cartilage.  The dermis is not as well vascularized as is that of other teleosts.  “Stressed” or ill free-ranging molas were typically partially predated by carnivores (California sea lions in Monterey Bay) who typically removed dorsal and anal fins, and scavengers (gulls) when the fish became immobilized. When performed, euthanasia methods were compliant with the most current version of the AVMA Guidelines, generally an overdose of MS-222. We have added the euthanasia protocol to the objectives in the introduction. In the Results section 3.3, we briefly describe each illness, injury and/or situation when euthanize was needed.

- Figure 2: The authors can´t link all sampling points with lines, as they measured cortisol at discrete times, not continuous. Moreover, since cortisol in teleost fish has a marked circadian rhythm, it is extremely important to show the time of day at which all samples were collected.

We believe that because these consecutive samples were taken over time, we can use a line graph. Additionally, we sampled mola between 11:00 and 13:00 to avoid any circadian rhythm issues.  We have added this to the Methods. 

 DISCUSSION

- Line 273: This is the first time the exact cortisol concentration per swab is shown in the manuscript. This information should be placed in the Results section.

These values are found within Results 3.3 along with Figure 2.

- Line 282: “seabream” instead of “sea bream”.

Corrected.

- Line 284: The results from the present study can´t be directly compared to those from the cited references as they measured the volume of collected mucus.

We have a statement that states this so that the readers know that we can’t directly compare the values. Additionally, you can’t directly compare cortisol values across species, such that you can’t state that one species is “more stressed” compared to another.

- Line 287: There is ample evidence that cephalopods do not produce cortisol… Please, revise the literature.

We have removed the cephalopod reference since in most cases authors haven’t conducted mass spec or HPLC to determine which specific steroid hormone/GC was measured.

- Lines 289-290: Please, refer to studies conducted in fish.

We have added the Lawrence et al 2018 reference here.

Reviewer 2 Report

This article titled preliminary investigation into developing the use of swabs for skin cortisol analysis for the Ocean Sunfish (Mola mola). The authors in their results show that the use of skin swab can be adopted to measure cortisol level as a maker of stress response. The article is relevant to the journal and has potential for high impact. I have the following concerns:

Concerns:

1.       Any idea if seasonal and temperature changes could impact on the cortisol levels?

2.       Is the cortisol release similar through out the whole body or different section of the body have different rates of cortisol secretion?

Author Response

We would like to thank you for your time and effort in our paper. Below we have added our responses to your questions in bold.

Concerns:

  1. Any idea if seasonal and temperature changes could impact on the cortisol levels?

This is an interesting question. Free-ranging mola are obviously subjected to the normal variation of sea surface temperature.  They are also known to make relatively deep dives, so they may be subjected to several degrees of temperature shift in short time periods when they drop below the thermocline.  However, in the aquarium, the temperature and lighting remain consistent all year.

  1. Is the cortisol release similar through out the whole body or different section of the body have different rates of cortisol secretion?

As mentioned in the discussion, this is a novel method, but skin mucus collection in Senegalese sole determined that cortisol was similar between the dorsal and ventral sides.

Reviewer 3 Report

animals-1871018 Preliminary Investigation into Developing the Use of Swabs for Skin Cortisol Analysis for the Ocean Sunfish (Mola mola)”.

GENERAL COMMENT:

The work entitled Preliminary Investigation into Developing the Use of Swabs for Skin Cortisol Analysis for the Ocean Sunfish (Mola mola) is a good work; the subject is original and of current and great interest.

In this study skin to study the stress physiology of the ocean sunfish (Mola mola) using skin mucus collected non-invasively using swabs. The objective was to validate this non-invasive method by taking opportunistic swabs throughout acclimatization and during stressful events.

Each individual used in the study was swabbed in three different body locations. Swabs were analyzed using a cortisol enzyme immunoassay and the absolute change of cortisol from baseline to during treatments and the different acclimation stages was examined.

The subject of the study is innovative, interesting and topical.

The introduction s is exhaustive.

Central argument is supported by evidence and analysis.

The methodology described by the author is accurate.

This work is a good work; it only needs some minor changes, for this reason I require minor revision.

DETAILED COMMENT:

·         Title

-The title is adequate.

·         Abstract

-The abstract is well structured and the objective of the study is clearly described.

Keywords are adequate but I suggest to change “fish” with “Mola mola” or “Ocean Sunfish”.

·         Introduction

The introduction section is exhaustive.

·         Materials and Methods

The section is well written and accurate.

·         Results

This section is accurate and detailed

·         Discussion

The discussion section is exhaustive and adequately discussed.

·         Tables and figures

Tables and Figures are clear and understandable.

·         References

The references are adequate.

Author Response

 We would like to thank you for your time and effort in our paper. Below we have added our responses to your questions in bold.

DETAILED COMMENT:

  • Title

-The title is adequate.

  • Abstract

-The abstract is well structured and the objective of the study is clearly described.

Keywords are adequate but I suggest to change “fish” with “Mola mola” or “Ocean Sunfish”.

We didn’t put those words in the keywords, because they are found in the title, which is searchable.

  • Introduction

The introduction section is exhaustive.

  • Materials and Methods

The section is well written and accurate.

  • Results

This section is accurate and detailed

  • Discussion

The discussion section is exhaustive and adequately discussed.

  • Tables and figures

Tables and Figures are clear and understandable.

  • References

The references are adequate.

Reviewer 4 Report

For this study the authors sought to validate the use of skin swabs for collecting and testing glucocorticoids in the skin mucous of the ocean sunfish (Mola mola). In terms of animal welfare, this method is an improvement over previous efforts to collect the mucous using cell scrapers, as the swabs are less likely to damage their integument. While this was not designed as an experimental study, but more of an opportunistic assessment of swabs taken when they could, they did evaluate GCs at various stressful times in the animals’ lives, such as when first acclimating to the aquarium, and during known incidents of illness and injury. Four of the six mola swabbed during the first week or month of captivity had GC concentrations which were 1.5x or more higher than baseline, and all 7 molas treated (or euthanized) for illness or injuries showed elevated GCs compared to baseline. Although baseline is somewhat questionably defined as the swabs taken at the time of initial exam, without information on if the mola had been handled for some time prior to this first swabbing – but even with that there were obvious increases in skin mucous GCs during events you would expect the fish to find stressful. The variability in where swabs were taken, and the lack of record of that info, is also problematic, as GC concentrations in skin secretions can vary greatly between body areas. It’s possible a more accurate measurement of baseline and more consistent swabbing locations would have produced even more obvious increases in GCs during the stressful events measured here. The study authors did a decent job detailing all of these issues, so the reader can decide how much weight to impart to these results.

Introduction

1.      Line 53: Change “is affected” to “has affected” or “is affecting”

2.      Lines 63-64: Please provide a citation.

Materials and Methods

1.      Line 144: Please define where/what the clavis is.

2.      Lines 167-172: Please provide additional detail about the EIA used to measure CORT, particularly as the manuscript cited for details (38) describes 3 different CORT EIA assays, none of them for skin swabs. In particular, what polyclonal antiserum did you use, what was the HRP bound to (I assume CORT?), and what sample dilutions did you use.

3.      Line 180: You could run statistical analysis comparing baseline samples to samples taken after 15 min of restraint.

4.      Line 182: What was used for baseline has not yet been defined.

Results

1.      Lines 191 and 192: Were the samples actually collected in triplicate (the same area swabbed 3 times), or are you referring to the 3 swabs taken from 3 different areas?

2.      Line 212: This is the first mention of how baseline was defined. This information should be in the Materials and Methods section. Also, could you include a justification of your choice of baseline measure? It is likely that at least some of the fish were not actually at baseline at the time of initial examination, as is illustrated in Fig. 1, with 3 of the fish appearing to have much lower CORT weeks later.

3.      Lines 217-239: For this whole paragraph, it is inconsistent which examples include the actual baseline and/or stressed CORT concentrations, and which only reference the fold change. You could also just put all this info in a table for easier reference.

4.      Lines 230-232: 1513.5 is only 10.5 fold higher than 143.7, not 38.1. Does the 38.1 refer to the CORT at the time of euthanasia, and the 1513.5 at the time of entrapment?

5.      Lines 233-234: The CORT concentrations listed here do not match the fold increase, I think they may have been swapped.

Discussion

1.      Lines 270-273: Another suggestion – At least some of the mola may be legitimately more stressed at the time of capture than when in captivity, due to increased food availability and/or decreased danger in captivity.

2.      Line 290: The lag time between a stressor and increased GC in blood plasma can be much longer in some poikilothermic species (see Tylan et al., 2020, Gen. Comp. Endocrinol., 287). In the Romero and Reed study they also didn’t see an increase in CORT within 3 minutes of capture in the one reptile species included in the study, and they didn’t have samples from later time points (and even some of the birds in the study did not show an increase within 3 minutes).

3.      Line 329: This might be a good point to include something about how standardizing where swabs are taken might improve reliability of the measurements.

Author Response

 We would like to thank you for your time and effort in our paper. Below we have added our responses to your questions in bold.

DETAILED COMMENT:

  • Title

-The title is adequate.

  • Abstract

-The abstract is well structured and the objective of the study is clearly described.

Keywords are adequate but I suggest to change “fish” with “Mola mola” or “Ocean Sunfish”.

We didn’t put those words in the keywords, because they are found in the title, which is searchable.

  • Introduction

The introduction section is exhaustive.

  • Materials and Methods

The section is well written and accurate.

  • Results

This section is accurate and detailed

  • Discussion

The discussion section is exhaustive and adequately discussed.

  • Tables and figures

Tables and Figures are clear and understandable.

  • References

The references are adequate.

 We would like to thank you for your time and effort in our paper. Below we have added our responses to your questions in bold.

Introduction

  1. Line 53: Change “is affected” to “has affected” or “is affecting”

Corrected

  1. Lines 63-64: Please provide a citation.

We have included the citation.

Materials and Methods

  1. Line 144: Please define where/what the clavis is.

The clavis is the truncated tail, which was defined in the Introduction when describing a mola.

  1. Lines 167-172: Please provide additional detail about the EIA used to measure CORT, particularly as the manuscript cited for details (38) describes 3 different CORT EIA assays, none of them for skin swabs. In particular, what polyclonal antiserum did you use, what was the HRP bound to (I assume CORT?), and what sample dilutions did you use.

The cortisol antibody # R4866 was provided along with whom produced it and the HRP. We have moved the antibody # next to “antiserum”.  The reference 38 is referring to the double-antibody EIA system we used. We have made that a new sentence to help clarify the details.

  1. Line 180: You could run statistical analysis comparing baseline samples to samples taken after 15 min of restraint.

Yes, another reviewer also suggested stats. So we have run some basic stats comparing within individuals (paired t-tests and1-way RM ANOVA). The statistical results are now in the Results section.

  1. Line 182: What was used for baseline has not yet been defined.

We agree that the word, baseline, is misleading so we have replaced it was “initial sample” or “post-acclimation sample” since we didn’t statistically find the baseline.

Results

  1. Lines 191 and 192: Were the samples actually collected in triplicate (the same area swabbed 3 times), or are you referring to the 3 swabs taken from 3 different areas?

We meant 3 swabs were taken from the same individual at the same time.  We have clarified this in the text.

  1. Line 212: This is the first mention of how baseline was defined. This information should be in the Materials and Methods section. Also, could you include a justification of your choice of baseline measure? It is likely that at least some of the fish were not actually at baseline at the time of initial examination, as is illustrated in Fig. 1, with 3 of the fish appearing to have much lower CORT weeks later.

As mentioned above, we have changed how we define that sample and removed the word baseline from the paper.

  1. Lines 217-239: For this whole paragraph, it is inconsistent which examples include the actual baseline and/or stressed CORT concentrations, and which only reference the fold change. You could also just put all this info in a table for easier reference.

We agree that we are providing a lot of information and a Table would organize it well. However, more than half of these data are in Figure 2. Therefore, we think it would be inappropriate to put the same information in a Table.

  1. Lines 230-232: 1513.5 is only 10.5 fold higher than 143.7, not 38.1. Does the 38.1 refer to the CORT at the time of euthanasia, and the 1513.5 at the time of entrapment?

Thank you for catching this error. Instead of 1513.5, it was supposed to be 5470.1 pg/mL.

  1. Lines 233-234: The CORT concentrations listed here do not match the fold increase, I think they may have been swapped.

Again, thank you for finding this error.  We have corrected them.

Discussion

  1. Lines 270-273: Another suggestion – At least some of the mola may be legitimately more stressed at the time of capture than when in captivity, due to increased food availability and/or decreased danger in captivity.

Thank you for this suggestion. We have added these to our explanation.

  1. Line 290: The lag time between a stressor and increased GC in blood plasma can be much longer in some poikilothermic species (see Tylan et al., 2020, Gen. Comp. Endocrinol., 287). In the Romero and Reed study they also didn’t see an increase in CORT within 3 minutes of capture in the one reptile species included in the study, and they didn’t have samples from later time points (and even some of the birds in the study did not show an increase within 3 minutes).

Thank you for this information. Another reviewer provided a reference for teleost fish species. So, we have removed this reference and used Lawrence et al. 2018 which is specific to fish.

  1. Line 329: This might be a good point to include something about how standardizing where swabs are taken might improve reliability of the measurements.

Thank you for this suggestion. It is mentioned above in the Discussion, but you are right, it should be emphasized again here.

Round 2

Reviewer 1 Report

Review of Santymire et al. "Preliminary investigation into developing the use of swabs for skin cortisol analysis for the ocean sunfish (Mola mola)".

Manuscript ID: ANIMALS-1871018 (v2)

The authors greatly improved the manuscript and thoroughly revised available bibliography. The information regarding the presence of glucocorticoid hormones of steroid nature in cephalopods is still controversial, but it´s up to the authors its inclusion in the text. The manuscript deserves to be published.